American Society for Microbiology | Microbiology Spectrum

# Expanding the microbiologist toolbox *via* new far-red-emitting dyes suitable for bacterial imaging

Massimiliano Lucidi,[1,2] Giulia Capecchi,[1] Daniela Visaggio,[1,2,3] Tecla Gasperi,[1] Miranda Parisi,[4] Gabriella Cincotti,[4] Giordano Rampioni,[1,3] Paolo Visca,[1,2,3] Kirill Kolmakov[5]

**ABSTRACT** Our understanding of bacteria is increasingly dependent on our ability to visualize cellular processes at the single-cell level with high spatial and temporal resolution. Advances in fluorescence microscopy are accompanied by an increasing demand for novel fluorophores that enable tagging specific bacterial components. In this context, new fluorescent probes emitting in the far-red (FR) are of particular interest as they reduce possible interference caused by sample autofluorescence and increase flexibility in multicolor imaging experiments. In this study, an extended set of previously reported and newly synthesized FR-emitting dyes has been characterized for their applicability in live single-cell imaging of the Gram-negative and Gram-positive prototype bacteria *Escherichia coli* and *Bacillus subtilis*. Toxicity tests demonstrated that these dyes do not interfere with the growth kinetics of both species, opening up the possibility of using them in live-cell imaging. Moreover, confocal laser-scanning microscopy imaging revealed that all the tested dyes can distinguish viable from dead bacterial cells. Among the newly synthesized fluorophores, the oxazine derivative KK 1905-NHS was particularly efficient in membrane staining and was effectively employed to monitor membrane biogenesis using a two-step labeling protocol on living cells. In addition, KK 1905-NHS was successfully used in super-resolution stimulated emission depletion microscopy. Overall, the new fluorophores presented in this study expand the microscopy toolbox, which is an asset for the investigation of fundamental bacterial processes.

**IMPORTANCE** By harnessing the versatility of fluorescence microscopy and super-resolution imaging, bacteriologists explore critical aspects of bacterial physiology and resolve bacterial structures sized beyond the light diffraction limit. These techniques are based on fluorophores with profitable photochemical and tagging properties. The paucity of available far-red (FR)-emitting dyes for bacterial imaging strongly limits the multicolor choice of bacteriologists, hindering the possibility of labeling multiple structures in a single experiment. The set of FR fluorophores characterized in this study expands the palette of dyes useful for microbiologists, as they can be used for bacterial LIVE/DEAD staining and for tagging the membranes of viable *Escherichia coli* and *Bacillus subtilis* cells. The absence of toxicity makes these dyes suitable for live-cell imaging and allows monitoring of bacterial membrane biogenesis. Moreover, a newly synthesized FR-fluorophore can be employed for imaging bacterial membranes with stimulated emission depletion microscopy, a super-resolution technique capable of increasing the resolving power of conventional microscopes.

**KEYWORDS** bacterial imaging, LIVE/DEAD staining, membrane biogenesis, fluorescent dyes, confocal microscopy, super-resolution microscopy

Address correspondence to Massimiliano Lucidi, massimiliano.lucidi@uniroma3.it, or Kirill Kolmakov, k.kolmakov@glyxera.com.

The authors declare no conflict of interest.

Ever since Antonie van Leeuwenhoek's groundbreaking work introduced us to *dierkeen*, little living beings often referred to as *animalcules*, microscopy techniques have remained a primary tool employed by scientists for exploring the microbial world. To date, techniques such as confocal and super-resolution microscopy are essential to visualize specific cellular components with high spatial and temporal resolution, an indispensable requirement for the understanding of key bacterial processes. Advances in fluorescence microscopy techniques are paralleled by the development of novel fluorophores with increased selectivity toward specific bacterial targets and improved photochemical properties (1).

Many innovations in the field of the generation of new fluorophores were achieved after the discovery of the green fluorescent protein (GFP) and its derivatives, providing a multicolor palette of dyes with different spectral properties (i.e., excitation and emission wavelengths), enhanced brightness, improved pH resistance, and good photostability (1, 2). However, although GFP and its variants are extremely useful for tracking the expression and localization of proteins in cells, their quantum yield is usually lower than that of organic dyes; they exhibit oligomerization tendency, fluorescence loss during fixation, and moderate cytotoxicity in overexpression conditions (2, 3). Besides the generation of novel variants of fluorescent proteins, the design of novel organic dyes is essential to expand the microscopist toolbox. Consequently, the request for small-molecule probes with less steric bulk and faster rates of labeling is increasing (1, 4).

Currently, many new organic dyes have been designed for different microscopy approaches. These dyes are based on various molecular scaffolds such as fluorescein, cyanine, oxazine, squaraine, BODIPY, xanthene, benzobisthiadiazole, and rhodamine (5–15). Many of the available fluorophores emit in the visible range, and their usage is hampered by sample autofluorescence, light absorption, and scattering, which reduce the signal-to-background ratio and contrast in fluorescence images (15). In addition, contrary to most fluorescent proteins, some organic dyes present high cytotoxicity, limiting the possibility of live-cell imaging (4), which is essential in bacteriology for gauging the physiological state of bacteria, untangling the intricate mechanisms at the basis of the bacterial divisome, performing real-time transcriptional analysis at the single-cell level, investigating phenotypic heterogeneity in bacterial populations, and exploring the host-bacteria interactions (4). Despite the significant array of applications, the fluorophores currently employed for bacterial staining are mainly confined to those outlined in Table S1.

Another central field of bacteriology is the ability to define if and how microbes survive after exposure to different environmental stressors (e.g., antibiotics) and within the host, and many protocols to assess bacterial viability rely on the use of fluorescent probes that can be employed to discriminate viable from membrane-damaged (i.e., dead) cells by using microscopy approaches (16–20). In this regard, propidium iodide (PI) is one of the most commonly used probes for the estimation of bacterial viability in fluorometric and microscopy approaches. PI is a cell-impermeable dye with a positive net charge that enters only cells with damaged membranes, binds to DNA, and proportionally emits red fluorescence (17, 21).

The resolving power of conventional fluorescence microscopy depends on the wavelength of the fluorescence light and constraints on both the lateral ($xy$ ~200–250 nm) and axial ($z$ ~500–600 nm) resolutions (4, 22). Different approaches have been proposed to overcome the diffraction limit, and various nanoscopic techniques have been successfully used to generate accurate images of bacterial cells, which themselves only marginally exceed the diffraction limit (4, 18, 23). Currently, super-resolution methods are not often used in microbiological laboratories because of the shortage of suitable dyes (1, 18, 23–25). Stimulated-emission depletion (STED) microscopy is a super-resolution technique able to increase the optical resolution of confocal laser scanning microscopy (CLSM) up to one order of magnitude (26). However, this resolution improvement comes at the cost of using high laser intensities, and it requires photostable dyes with high fluorescence quantum yields capable of resisting photobleaching

(26). This limitation restricted the use of STED microscopy in bacteriology to visualizing bacterial division machinery, membrane microdomains, and cytoskeletons, primarily in *Escherichia coli* and *Bacillus subtilis* (18, 23).

A previous comparative study investigated the STED applications of red-emitting rhodamine dyes, demonstrating that fluorophore functionalization with charged polar functional groups (sulfate, phosphate) was crucial for the dye's performance (9). However, although a large set of dyes was tested, positively charged compounds were not considered. To bridge this gap, we developed and tested new water-soluble far-red (FR) fluorophores with zero and positive net charges. As solubilizing groups, oxyalkyl, alkylamino (i.e., piperazine), carboxyethyl, and/or carboxyl groups were considered. The new custom-made dyes, together with a set of previously synthesized dyes with various auxiliary groups, were tested on *B. subtilis* and *E. coli*.

## RESULTS AND DISCUSSION

### The tested FR-emitting dyes are suitable for LIVE/DEAD staining of bacterial cells

Fluorescence microscopy and its super-resolution nanoscopy derivative approaches demand markers with red and/or FR excitation and emission bands, characterized by high quantum yield, elevated photostability, and high solubility in water, to facilitate biological sample manipulation during staining and avoid unspecific labeling (5, 15). Moreover, the availability of lipophilic and hydrophilic derivatives of the same chromophore is desirable because it provides additional flexibility in labeling biological targets presenting different polarities. Hydrophilicity would be advantageous for labeling polar substances such as lipid head groups, while lipophilic derivatives are useful for labeling nonpolar domains (e.g., lipid acyl chains) (5, 15).

To expand the palette of FR-emitting dyes available for bacterial labeling, an *N*-hydroxysuccinimidyl (NHS)-containing oxazine (namely, KK 1905-NHS) and two rhodamines (namely, KK 1116 and KK 1518) have been generated, and their performances were compared to a previously synthesized set of dyes never applied for bacterial imaging before (i.e., KK 114S, KK 1115, KK 1517, KK 1517-NHS, KK 1558-NHS, KK 1905, and STAR RED in its deactivated form) or previously employed for STED bacterial imaging (i.e., KK 114-NHS and/or STAR RED-NHS) (18). Figure 1 displays the chemical structure of the fluorophores used in this work.

As a preliminary characterization of the fluorophores, their relative quantum yield ($\Phi_f$) and absorption/emission maxima were determined (Table 1). PI was included in the analysis as a reference dye. All the fluorophores presented comparable $\Phi_f$ values, ranging from 50% to 62% relative to the Oxazine 170 dye, except for KK 1558-NHS ($\Phi_f = 17\%$), whose $\Phi_f$ was similar to PI ($\Phi_f = 20\%$) (Table 1). As expected, all the fluorophores displayed maximum emission in the FR bandwidth. In particular, the two-oxazine derivatives KK 1905 and KK 1905-NHS presented an emission maximum of approximately 25 and 65 nm shifted to FR wavelengths compared to the other rhodamine and oxazine derivatives and PI, respectively (Table 1).

The ability of the rhodamine and oxazine derivatives to stain bacterial cells was tested on *B. subtilis* and *E. coli*, considered prototypic organisms of Gram-positive and Gram-negative species, respectively. Except for the rhodamines KK 1517, KK 1518 (purposefully synthesized with a net charge of +2), and the oxazine KK 1905-NHS, all the dyes largely failed in staining *B. subtilis* (Fig. 2) and *E. coli* (Fig. 3) cells, with only a few *E. coli* cells producing detectable signals in the presence of KK 114-NHS, KK 114S, KK 1115, KK 1116, KK 1517-NHS, KK 1558-NHS, KK 1905, and STAR RED. Interestingly, the few stained *E. coli* cells were characterized by small dimensions, irregular edges, phase contrast decrease, and/or wrinkled membranes in the differential interference contrast (DIC) images, while appearing brighter and stained in the cytoplasm in the fluorescent acquisition channel (Fig. 3). These characteristics conform to morphological markers previously described for dead bacterial cells (28, 29) and suggest that the tested dyes would selectively stain the cytoplasm of cells with damaged membranes, as previously reported for PI (21).

**TABLE 1** Properties and staining behavior of different FR-emitting dyes

| Dye | Net charge | Maximum absorption/emission wavelength (nm) | $\Phi_f$ (%)[a] | Cytoplasm labeling in dead cells | Membrane labeling in untreated cells | Reference[b] |
|---|---|---|---|---|---|---|
| KK 114S | −4 | 635/652 | 54 | + | − | 9 |
| KK 114-NHS | −1 | 634/654 | 62 | + | − | 7 |
| KK 1115 | −1 | 635/655 | 58 | + | − | 7 |
| KK 1116 | 0 | 637/656 | 60 | + | − | This work |
| KK 1517 | 0 | 639/658 | 52 | + | +/− | 9 |
| KK 1517-NHS | +1 | 640/661 | 50 | + | − | 9 |
| KK 1518 | +2 | 638/657 | 56 | + | + | This work |
| KK 1558-NHS | −3 | 634/654 | 17 | + | − | 9 |
| KK 1905 | +1 | 661/680 | 53 | + | − | 8 |
| KK 1905-NHS | +1 | 662/680 | 50 | + | + | This work |
| STAR RED[c] | −1 | 635/655 | 59 | + | − | 8 |
| PI[d] | +2 | 535/615 | 20 | + | − | 21 |

[a]Relative quantum yields measured in PBS buffer at pH = 7.4, using oxazine 170 as a reference dye (9).
[b]See Fig. 1 for structures.
[c]Unreactive form of the dye [see reference (27) and compound 4a therein].
[d]Properties refer to the DNA-binding state of the dye.

To investigate this possibility, *B. subtilis* and *E. coli* bacterial suspensions were heat-inactivated before staining. Complete cell killing was verified by colony-forming unit (CFU) counts on agar plates. As observed for the reference dye PI, both *B. subtilis* and *E. coli* cells resulted in homogeneous staining in the cytoplasm and appeared brighter than the untreated viable samples when treated with the tested FR dyes (compare the left and right panels of Fig. 2 and 3).

Overall, these data indicate that, regardless of their molecular structure and net charge, the rhodamine and oxazine derivatives KK 114S, KK 114-NHS, KK 1115, KK 1116, 1517-NHS, 1558-NHS, KK 1905, and STAR RED are all able to selectively stain dead cells, hence representing valuable alternatives to PI for discriminating live and dead bacterial cells.

As mentioned above, only KK 1517, KK 1518, and KK 1905-NHS could stain viable bacteria. In particular, the rhodamine KK 1517 was able to decorate the cell periphery of all the *B. subtilis* cells (Fig. 2) and of some *E. coli* cells (Fig. 3), partially penetrating the cytoplasm of both species. On the other hand, KK 1518 and KK 1905-NHS selectively stained the cell periphery of *B. subtilis* and *E. coli* cells, with the latter generating brighter and more contrasted images (Fig. 2 and 3).

To verify that KK 1518 and KK 1905-NHS decorate the cell periphery without massively penetrating bacterial cytoplasm unless the bacteria were dead, these labels were used in combination with other dyes usually employed for LIVE/DEAD staining, such as PI and SYTO 9 (17, 21). SYTO 9 is a cell-permeant fluorophore that labels bacterial DNA, emitting green fluorescence. Its emission is quenched by the red-emitting PI by fluorescence resonance energy transfer, resulting in green- and red-tagging of viable and dead bacteria, respectively (17, 21). Therefore, *B. subtilis* cells harvested from the death phase were simultaneously stained with SYTO 9, PI, and KK 1518 or KK 1905-NHS. Coherently with our hypothesis, this multicolor labeling strategy resulted in the DNA staining of viable bacteria with SYTO 9 and cellular envelope tagging with KK 1518 or KK 1905-NHS, while dead bacterial cells were tagged with both PI and KK 1518 or KK 1905-NHS, without undesired interference or bleed-through artifacts (Fig. S1).

## Investigation of the molecular targets of the oxazine and rhodamine labels

To shed light on their molecular target(s), all the tested fluorophores were used to stain DNA, bovine serum albumin (BSA), and 1-palmitoyl-2-oleoyl-sn-glycero-3-phosphocholine (POPC)-based liposomes. For each dye, fluorescence fold-induction (FFI) was calculated as the ratio between the fluorescence of the dye-macromolecule complex and

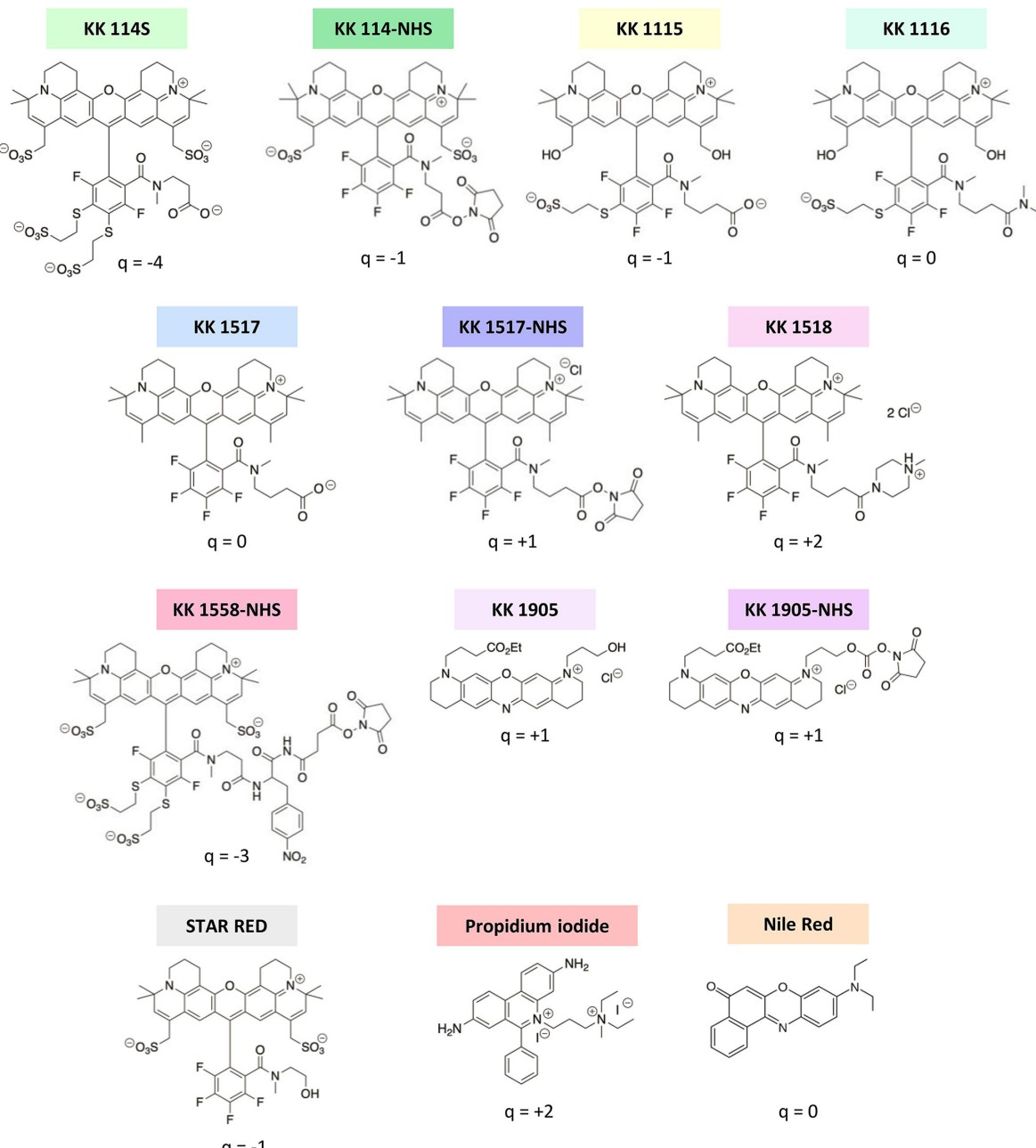

**FIG 1** Chemical structures of the fluorescent dyes used in this work. For each molecule, the net charge (*q*) is indicated.

dye autofluorescence in distilled water. FFI values >20% were arbitrarily chosen as predictors of reliable molecular interactions. Interestingly, all the oxazine and rhodamine derivatives used in this work were able to stain DNA and/or BSA to different extents (Fig. S2). The negatively charged compounds KK 114S, KK 114-NHS, KK 1115, KK 1558 NHS, and STAR RED stained BSA but not DNA, probably due to charge repulsion. Moreover, FFI values of KK 1517, KK 1518, and KK 1905-NHS in the presence of the POPC-based liposomes were comparable to the FFI value measured for the phospholipid-binding oxazine Nile Red (NR) (30), used as a control (Fig. S2). Intriguingly, NR FFI increased in the presence of BSA and, to a lesser extent, DNA (Fig. S2), raising the hypothesis that this molecule can also enter dead cells and stain their cytoplasm by non-selectively binding nucleic acids and proteins. To test this hypothesis, heat-killed *B. subtilis* cells were stained

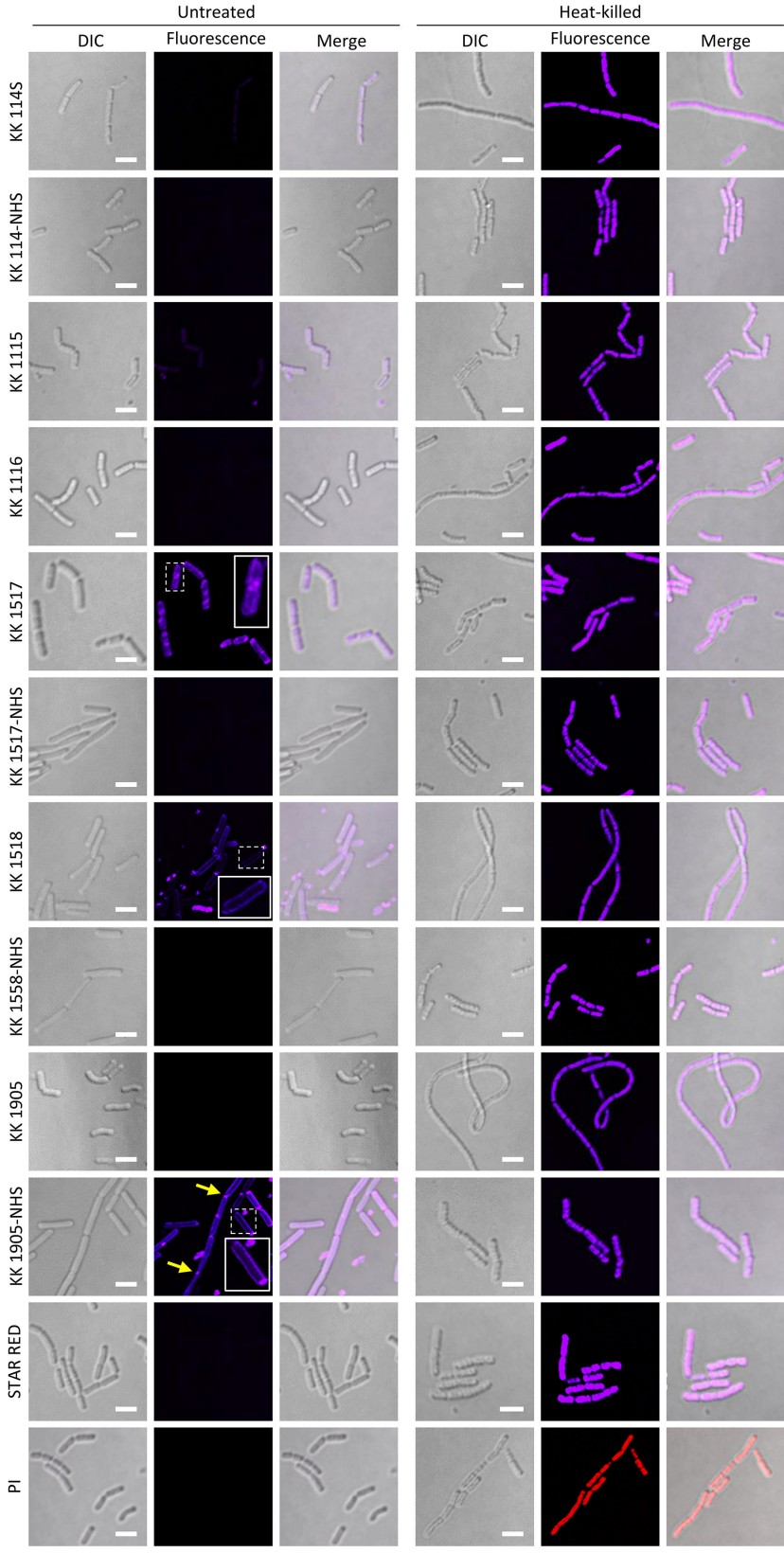

**FIG 2** *B. subtilis* cells stained with the indicated fluorescent dyes. Bacterial cells were inoculated in Luria-Bertani broth, harvested after 6 h of incubation at 37°C with shaking (250 rpm), washed with sterile saline, and diluted to $OD_{600} = 1$. An aliquot was killed by heating at 100°C for 30 min. Twenty microliters

**FIG 2** (Continued)

of the stained suspensions was spotted on a glass slide covered with 0.5% (wt/vol) agarose and visualized using the Nikon A1+ CLSM equipped with a 100× oil immersion objective (NA 1.49). For each dye, fluorescence, differential interference contrast (DIC), and fluorescence-DIC merged channels are shown. Scale bar: 2.5 µm. Insets represent an enlargement of the boxed region to highlight membrane labeling. Yellow arrows indicate labeled division Septa.

with NR in combination with KK 1518 or KK 1905-NHS before visual inspection by CLSM. As a result, all the cells appeared brighter and more homogeneously stained (Fig. S3), demonstrating that NR shares the same dead cell-tagging capability as our oxazine and rhodamine molecules.

These results indicate that when inside membrane-damaged cells, our dyes could bind nucleic acids, proteins, and/or hydrophobic molecules or cellular structures (e.g., lipid droplets), with consequent fluorescence emission whose intensity can vary depending on the dye-macromolecule interaction pair.

The high FFI values of KK 1518 and KK 1905-NHS in the presence of POPC-based liposomes (Fig. S2), together with their ability to decorate the cell periphery of *B. subtilis* (Fig. 2) and *E. coli* cells (Fig. 3), open the possibility of using these dyes as membrane-labeling compounds. To quantitatively assess the membrane labeling selectivity of KK 1518 and KK 1905-NHS, a colocalization experiment of *B. subtilis* cells stained with these compounds and NR was performed (Fig. 4a). The non-selective membrane-staining dye KK 1517 (Fig. 2 and 3) was included in this analysis as a control (Fig. 4). Through tracing arbitrarily drawn lines (depicted as dotted lines in Fig. 4a), the spatial resolution of NR and KK1517, KK 1518, or KK 1905-NHS was assessed by plotting fluorescence intensity values for each fluorophore against the line length. Leveraging NR's selectivity in labeling bacterial membranes (30), each maximum in the NR spatial resolution plots pinpointed a bacterial membrane (as denoted by black arrows in Fig. 4b). The extent to which the curves of KK1517, KK 1518, and KK 1905-NHS overlapped with NR's curves provided a qualitative estimation of the selectivity of each fluorophore for binding to the bacterial membrane. Notably, only the KK 1905-NHS spatial resolution plot was similar to that of NR, suggesting colocalization of these two dyes on bacterial membranes (Fig. 4b). This result was confirmed by the cytofluorograms (Fig. 4c). Accordingly, the higher values of colocalization area and Pearson and Manders coefficients between KK 1905-NHS and NR, compared to those between KK1517, KK 1518, and NR (Fig. 4d), proved that KK 1905-NHS can tag bacterial membranes with the highest selectivity among the tested dyes.

Interestingly, the membrane labeling property of these three compounds is charge-independent, as KK 1517, KK 1518, and KK 1905-NHS have a net charge of 0, +2, and +1, respectively. In this respect, none of the tested dyes with a negative net charge were functional in membrane labeling. On the other hand, the presence of a reactive NHS group does not seem to be predictive of bacterial envelope staining. Indeed, KK 1905-NHS and KK 1517-NHS are both endowed with an NHS moiety but exhibit different capabilities to stain the bacterial envelope.

## The assayed FR dyes are non-toxic to bacteria and can be used in live-cell imaging

Fluorescent molecules convert only a fraction of absorbed energy into fluorescence, thus requiring appropriate laser intensities to obtain a good contrast in fluorescence microscopy images. Therefore, fluorophores with high quantum yields are needed to reduce the excitation energy input and minimize phototoxicity, allowing live cell imaging experiments to be performed (31). To evaluate the possibility of using the dyes for time-lapse live-cell imaging, their toxicity was preliminarily tested. Briefly, *B. subtilis* and *E. coli* cells were incubated in Luria-Bertani broth (LB) supplemented with 10 µM of the fluorophores, and the bacterial growth was monitored over time. No variations in growth rates were observed irrespective of the presence or absence of the fluorophores (Fig. S4),

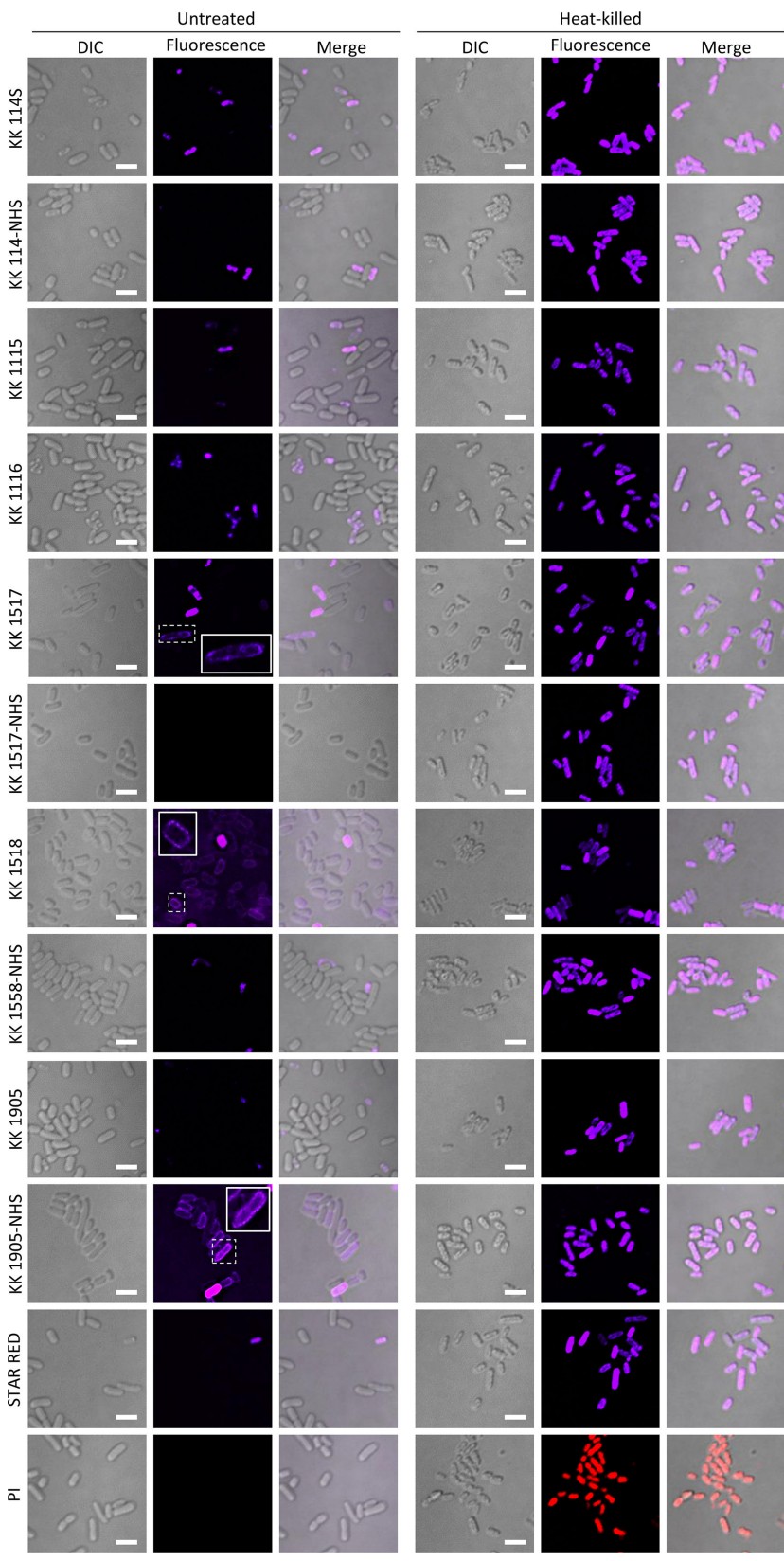

**FIG 3** *E. coli* cells stained with the indicated fluorescent dyes. Bacterial cells were inoculated in Luria-Bertani broth, harvested after 6 h of incubation at 37°C with shaking (250 rpm), washed with sterile saline, and diluted to $OD_{600} = 1$. An aliquot was killed by heating at 100°C for 30 min. Twenty

**FIG 3** (Continued)

microliters of the stained suspensions was spotted on a glass slide covered with 0.5% (wt/vol) agarose and visualized using the Nikon A1+ CLSM equipped with a 100× oil immersion objective (NA 1.49). For each dye, fluorescence, differential interference contrast (DIC), and fluorescence-DIC merged channels are shown. Scale bar: 2.5 µm. Insets represent an enlargement of the boxed region to highlight membrane labeling.

indicating that the tested dyes are not toxic for Gram-positive and Gram-negative bacteria, encouraging their use in live-cell imaging experiments.

Since no dye of the whole set was found to be toxic *per se*, KK 1905 was used in a time-lapse experiment as a representative of the other dyes, preferentially staining *B. subtilis* dead cells (Fig. 2). Briefly, *B. subtilis* cells harvested from the death phase were spread on an LB 0.5% (wt/vol) agarose pad supplemented with 10 µM of KK 1905, and cell replication was followed by CLSM imaging. Video S1 displays that only unlabeled cells were able to increase their cellular volume and undergo binary fission, while KK 1905-tagged cells were not. This result confirmed that KK 1905 (i) does not interfere with bacterial replication, (ii) does not stain healthy cells, and (iii) is able to stain the cytoplasm of dead bacteria.

Overall, these data indicate that the rhodamine and oxazine derivatives KK 114S, KK 114-NHS, KK 1115, KK 1116, 1517-NHS, 1558-NHS, KK 1905, and STAR RED are valuable dyes to discriminate viable from dead bacteria in live-cell imaging experiments.

## Monitoring membrane biogenesis using the oxazine KK 1905-NHS

The study of membrane biogenesis is currently a hot research topic in bacteriology due to the large interest related to the mechanisms behind cell replication machinery (32), cell permeability to antibiotics (33), solute transport (34), analysis of bacterial subpopulations (35), and production of extracellular vesicles (36). At present, the most employed dyes for bacterial membrane labeling are the phospholipid-binding oxazine NR (30, 37–40) and the pyridinium dibromide FM 4–64 (41–44), both presenting a red emission bandwidth. Several less-used, commercially unavailable dyes for bacterial membrane labeling have also been described [reviewed by Yoon and coworkers (45)]. The lack of near-infrared (IR)-emitting fluorophores for membrane labeling of bacterial cells limits the possibility of studying membrane biogenesis *via* multicolor imaging (23, 46). The FR emission of KK 1905-NHS, which labels bacterial membranes (Fig. 2 and 3, and S1), makes it suitable, at least in principle, for the combined use of other membrane labeling dyes (e.g., NR) to investigate membrane biogenesis.

To explore the possibility of using KK 1905-NHS for membrane labeling in live-cell imaging, *B. subtilis* cells were stained with this fluorophore, spread onto a dye-free LB 0.5% (wt/vol) agarose pad, and imaged during growth in a CLSM-based time-lapse experiment. As shown in Video S2, cells with cytoplasmic staining did not replicate, in line with the evidence that dead cells are fully stained by KK 1905-NHS (Fig. 2). Conversely, the fluorescent signal associated with the membrane of replicating cells decreased during time in parallel to the increase of their cellular volume, indicating that the "old" membranes tagged with the KK 1905-NHS dye were substituted by newly synthesized unlabeled membranes during bacterial growth.

To investigate the feasibility of using KK 1905-NHS and NR in combination, their whole-cell emission spectra were preliminarily determined together with those of other dyes described in this study. When labeling dead *E. coli* cells, all the fluorophores showed similar emission spectra, with an emission peak at approximately 660 nm, except for KK 1905 and KK 1905-NHS, which are shifted to the FR with a peak at approximately 680 nm (Fig. S5). These results corroborate the data shown in Table 1 and demonstrate that binding to their bacterial targets does not cause undesired bathochromic shifts. Notably, in the same conditions, NR presented an emission maximum at approximately 625 nm (Fig. S5). In line with this evidence, KK 1905-NHS and NR were successfully used in combination to stain liposomes without bleed-through artifacts (Fig. S6).

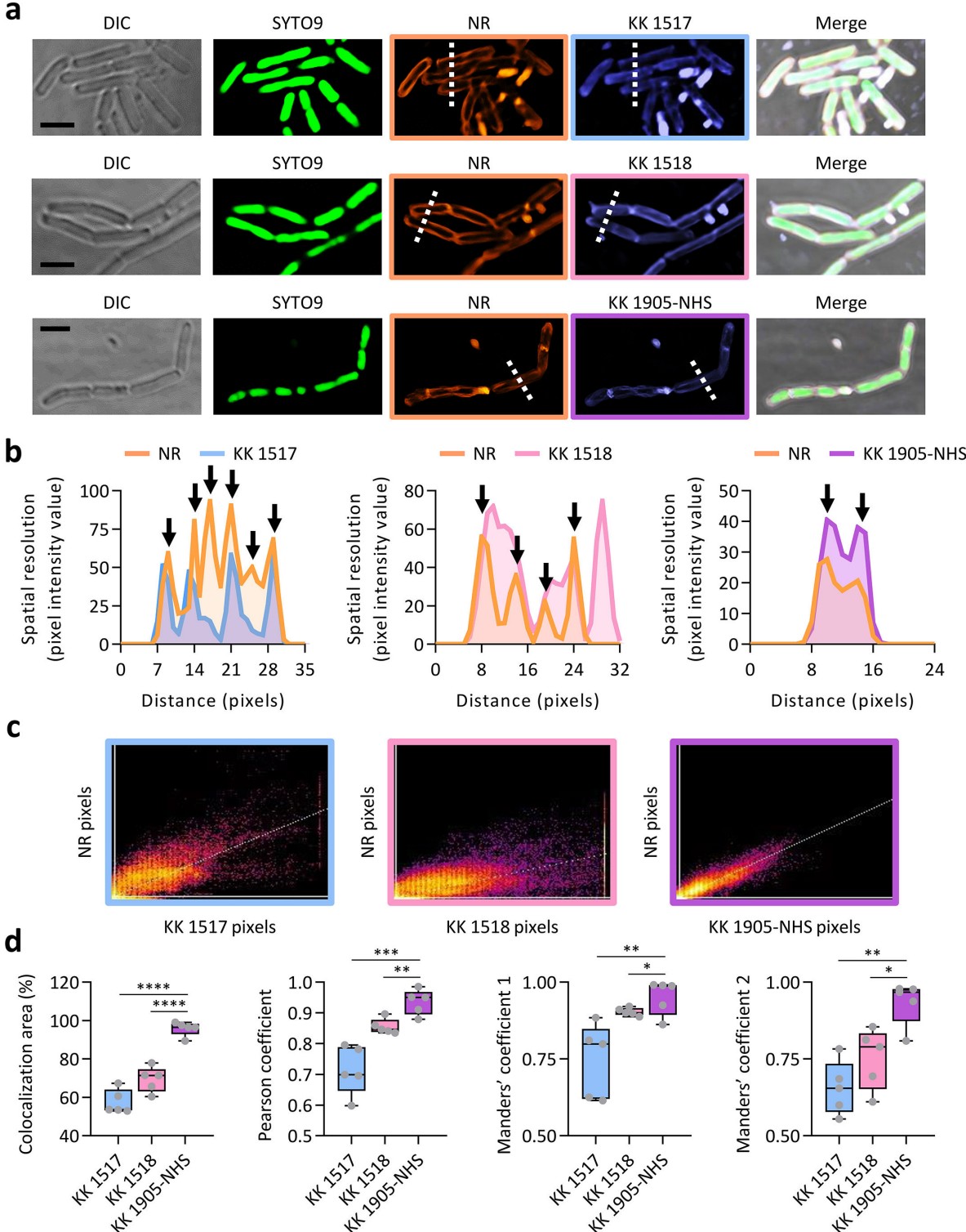

**FIG 4** Colocalization of KK 1517, KK 1518, and KK 1905-NHS with the membrane labeling fluorophore NR. (a) Representative images of *B. subtilis* cells grown in Luria-Bertani broth for 4 h and stained with SYTO 9, NR, and the indicated fluorophores. Scale bar: 2.5 μm. Dotted lines in the NR, KK 1517, KK 1518, and KK 1905-NHS fluorescence channels indicate the region of interest used to plot the spatial resolution. (b) Spatial resolution plots. Black arrows indicate the membranes corresponding to the relative maximum values of the NR spatial resolution plots. (c) Cytofluorograms of the shown NR and KK 1517, KK 1518, or KK 1905-NHS merged images. (d) Colocalization parameters between NR and the indicated fluorophores calculated on five CLSM images. The limits of each box represent the first and third quartiles, and the values outside the boxes represent the maximum and minimum values. The line dividing the box indicates

**FIG 4** (Continued)

the median value. Asterisks indicate statistically significant differences between KK 1905-NHS, KK1517, and KK 1518 (*$P < 0.05$; **$P < 0.01$; ***$P < 0.001$; ****$P < 0.0001$).

A two-step labeling method was implemented to analyze the turnover of bacterial membranes by using the combination of KK 1905-NHS and NR in synchronously dividing *B. subtilis* cells. Briefly, *B. subtilis* cells were stained with NR and incubated in LB at 37°C for up to 4 h. During incubation, bacteria were harvested at different time points (0, 0.5, 1, 2, and 4 h), and membranes were counterstained with KK 1905-NHS. Then, the contribution of membrane labeling for NR and KK 1905-NHS was calculated for a total of $n = 50$ cells for each time point (Fig. 5a). As expected, NR fluorescence decreased over time, suggesting that the old phospholipids were progressively replaced with newer ones (Fig. 5b and c). Similar results were obtained when *B. subtilis* cells were stained with KK 1905-NHS before incubation in LB and then counterstained with NR (Fig. S7), indicating that the decrease of NR labeling over time (Fig. 5c) was not due to the fluorophore washing or spontaneous diffusion into the LB medium.

Taken together, these results represent a proof of concept demonstrating that KK 1905-NHS and NR can be successfully employed for monitoring bacterial membrane turnover in time-lapse experiments.

## STED nanoscopy of bacteria labeled with the oxazine KK 1905-NHS

STED is a CLSM derivative super-resolution technique capable of pushing the resolution limits beyond the diffraction barrier (25). STED microscopy overcomes the diffraction limit by reversibly silencing (depleting) fluorophores at predefined positions in the diffraction-limited excitation regions. Photon emission only occurs from the active fluorophores in the corresponding areas, enabling the distinction of features closer than the diffraction limit. This fluorescent confinement is obtained by the alignment of both the CLSM excitation beam and a second beam, called the STED beam (25). Consequently, the simultaneous use of two coaligned laser sources necessitates the use of photostable dyes resistant to photobleaching for effective STED imaging (25).

The oxazine derivatives have been proven to be highly photostable and suitable for many super-resolution applications (40); therefore, we investigated the potential of the oxazine KK 1905-NHS as a fluorescent marker for STED nanoscopy. For this purpose, *B. subtilis* cells were stained with KK 1905-NHS before visualization with CLSM and STED microscopy. Figure 6a shows a resolution enhancement in STED images compared to the images collected with CLSM before and after deconvolution. To quantitatively evaluate the resolution improvement, we plot profile lines along three different regions of interest (ROIs) to depict the spatial resolution (Fig. 6b). Because different photon counts were recorded in CLSM and STED images, the profile lines plotted along different ROIs were normalized. The profile plots for both raw CLSM and STED images were also smoothed using a second-order polynomial to average five neighboring pixels, as previously described (18). The STED resolution enhancement can be inferred from the full width half maximum (FWHM) values of the profile lines presented in Table 2.

Overall, these data indicate that KK 1905-NHS is a potentially useful dye for STED super-resolution microscopy. It is noteworthy that the KK 1905-NHS dye has no polar groups with negative charges, which were previously considered to be crucial for a good

**TABLE 2** FWHM values for selected ROIs on CLSM and STED images of KK 1905-NHS-stained *B. subtilis* cells

| ROI[a] | Raw CLSM (µm) | Raw STED (µm) | Deconvoluted CLSM (µm) | Deconvoluted STED (µm) | Raw STED/CLSM FWHM ratio (µm) | Deconvoluted STED/CLSM FWHM ratio (µm) |
|---|---|---|---|---|---|---|
| 1 | 0.48 | 0.17 | 0.45 | 0.15 | 0.35 | 0.33 |
| 2 | 0.45 | 0.35 | 0.43 | 0.26 | 0.77 | 0.60 |
| 3 | 0.54 | 0.40 | 0.32 | 0.29 | 0.74 | 0.91 |

[a]The considered ROIs are depicted in Fig. 6a.

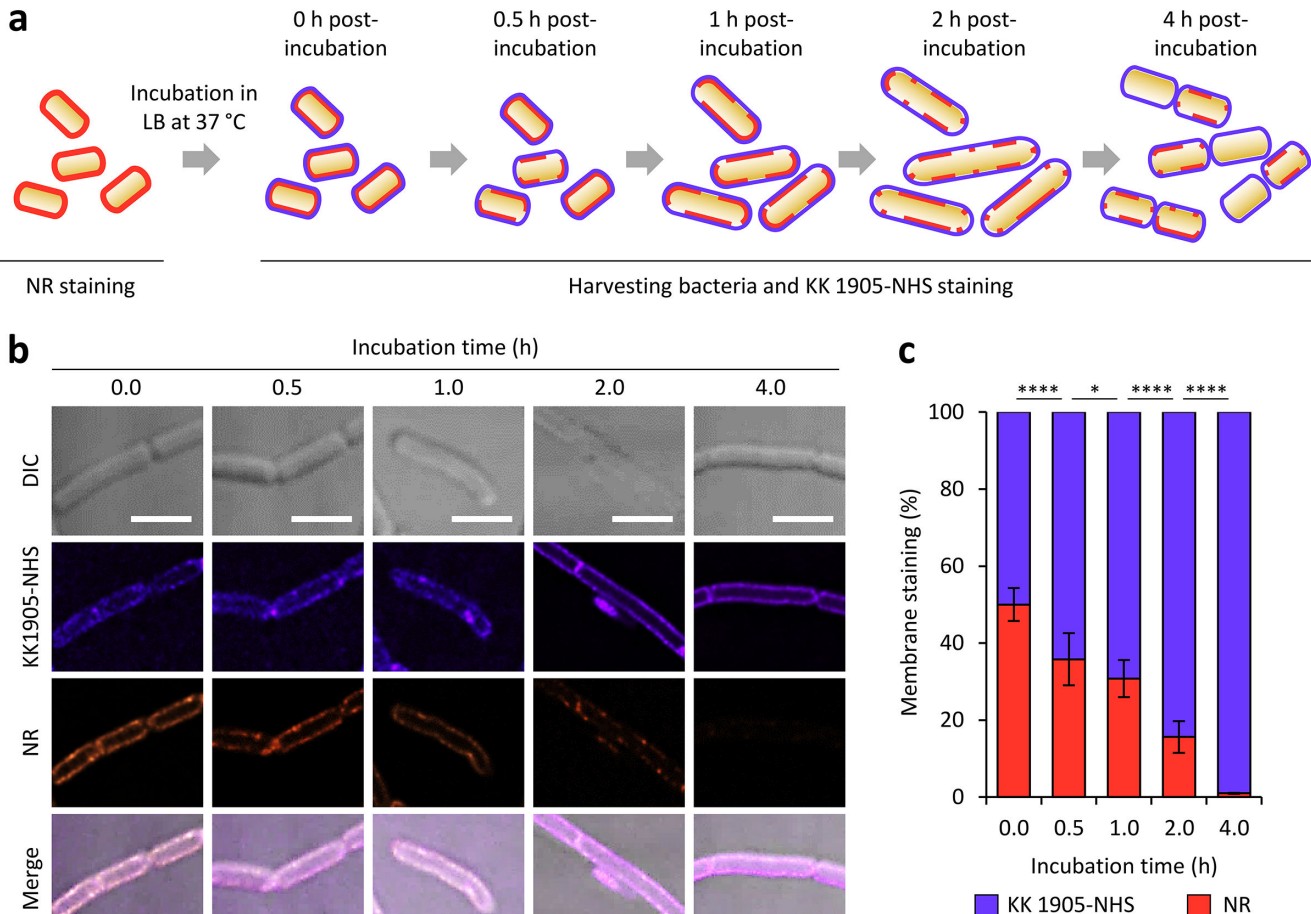

**FIG 5** Monitoring bacterial membrane biogenesis with two-step labeling. (a) Schematic representation of the two-step labeling procedure. *B. subtilis* cells were stained with NR and incubated at 37°C in LB. At 0, 0.5, 1, 2, and 4 h post incubation, bacteria were harvested, stained with KK 1905-NHS, and visually inspected using CLSM. Orange and violet cell outlines indicate the NR-labeled "old" membranes and the KK 1905-NHS-labeled membranes, respectively. (b) Representative images of bacterial membranes obtained using the Nikon A1+ CLSM equipped with a 100× oil immersion objective. Scale bar: 2.5 µm. (c) For the indicated time points, NR and KK 1905-NHS-associated fluorescence were quantified for different bacterial cells ($n = 50$). Asterisks indicate statistically significant differences in NR and KK 1905-NHS fluorescence percentages in the analyzed cell membranes between the indicated time points (*$P < 0.05$; ****$P < 0.0001$).

STED performance (9). Even though these preliminary data support KK 1905-NHS's compatibility with STED imaging, further studies should be conducted to compare the performances of this oxazine with a panel of fluorophores routinely employed for STED imaging of prokaryotes.

## Conclusions

Microbiologists have grappled with the challenge of visualizing bacterial structures since the early days of brightfield microscopy. Despite the historical significance of diffraction-limited, label-free techniques (e.g., phase contrast and DIC), their effectiveness has been hampered by the inability to label specific bacterial subcellular components, thus limiting their role as supportive techniques for label-based ones (23, 47). Recent progress has effectively overcome the diffraction barrier, enabling the examination of bacterial cells at the nanoscale without requiring extensive sample manipulation or labeling. This accomplishment is notably demonstrated through the widespread utilization of atomic force microscopy and its derivatives in bacteriology (47–51). Nevertheless, various label-free techniques have been integrated with diverse label-based microscopies to glean supplementary information (18, 51–54). Within this framework, the significance of having

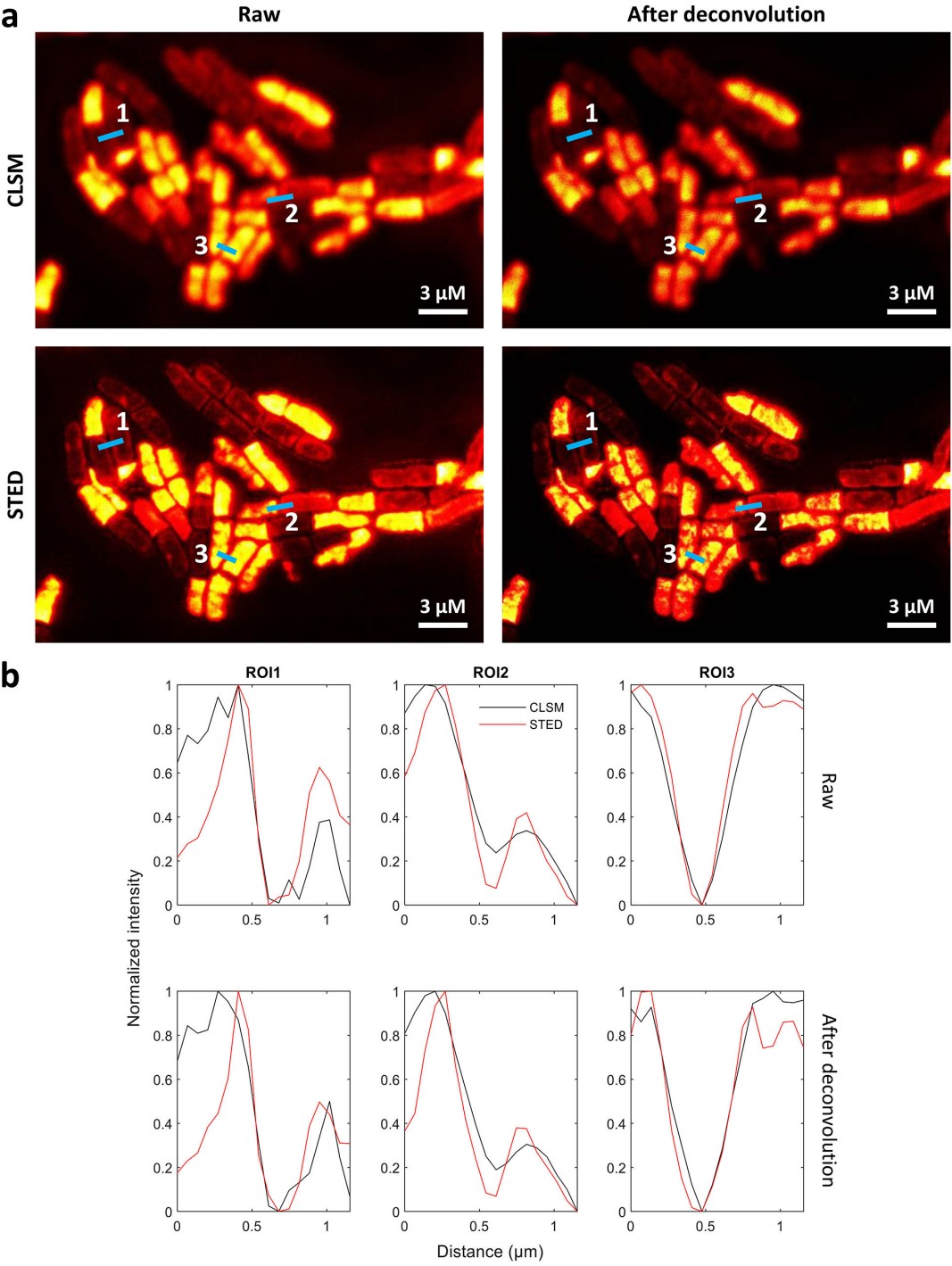

**FIG 6** STED imaging of *B. subtilis* cells using KK 1905-NHS. (a) Comparison between CLSM and STED images of *B. subtilis* cells harvested from the dead phase and stained with KK 1905-NHS before and after deconvolution. (b) Profile lines along three relevant ROIs quantitatively showcase the resolution improvement; the corresponding FWHM values are reported in Table 2.

dyes for fluorescent imaging cannot be overstated, as they play a pivotal role in staining specific bacterial components. In this context, FR fluorescent probes have emerged in recent years as valuable tools to complement other fluorescent molecules emitting in the visible range (15). The availability of multifunctional FR fluorophores allows the expansion of the multicolor palette of bacteriologists.

Here, we explored the applications of a set of FR-emitting rhodamine and oxazine dyes for *E. coli* and *B. subtilis* imaging. All the dyes feature a bright fluorescence accompanied by good quantum yields and water solubility. Although the fluorophores have different net charges (from −4 to +2) and are functionalized with different groups, all of them can be used to tag dead bacterial cells, analogously to PI (21), thus expanding the availability of multicolor dyes for LIVE/DEAD applications.

The presented dyes can be potentially used in combination with other cell-permeant molecules, such as SYTO 9 (17, 20, 21), 4′,6-diamidino-2-phenylindole (DAPI) (55), and BODIPY derivatives (56), to mark all the cells regardless of their viability and to give an estimation of the number of viable and dead bacteria, opening the possibility to evaluate bacterial responses to environmental stressors, such as antibiotic treatments (20, 57). Moreover, toxicity tests on *B. subtilis* and *E. coli* revealed that all the fluorophores did not interfere with bacterial replication, allowing their usage in live-cell imaging experiments, as substantiated by the time-lapse approaches employed in this work.

Of the whole set of tested fluorophores, only three were able to stain the bacterial membranes (although with different selectivity), i.e., KK 1517, KK 1518, and KK 1905-NHS, having a net charge of 0, +2, and +1, respectively. None of the negatively charged compounds worked, probably due to charge repulsion. Indeed, the outer membrane of Gram-negative bacteria and the cytoplasmic membrane/cell wall of Gram-positive bacteria are decorated with lipopolysaccharide and teichoic acids, respectively, both making their outer shells negatively charged (58, 59). However, apart from the net charge, no direct correlation between the chemical structure of the dyes and their capability to label bacterial components was observed. For instance, KK 1905 and KK 1905-NHS share the oxazine scaffold (like NR) and have the same net charge, but only the latter dye was able to stain bacterial membranes. On the other hand, KK 1517 and KK 1518 stain bacterial membranes, yet to a lesser extent. Remarkably, KK 1116, a model compound that has a zero net charge like KK 1517 and belongs to the same rhodamine family, showed no staining, demonstrating that the ionizable (anionogenic) sulfonic acid group ($SO_3H$) is prohibitive for label bacterial components, at least in live bacteria. In addition, the attachment of a positively charged ligand (*N*-methylpiperazine) to compound KK 1518 dramatically expanded the range of its application. This straightforward methodology can likely be extended to dyes of other classes.

In addition to the ability to label bacterial membranes, the FR-shifted emission spectrum of KK 1905-NHS makes this oxazine dye suitable for the investigation of membrane biogenesis. This is also possible in combination with other red-emitting phospholipid-binding dyes, such as NR. Moreover, the KK 1905-NHS photostability and non-toxicity allow imaging of bacterial membranes both with optical diffraction-limited approaches (e.g., CLSM) and by means of super-resolution techniques (e.g., STED microscopy).

Summarizing, the present work describes the staining behavior of a set of fluorescent compounds with potentially very useful properties. These fluorophores could be exploited by the scientific community not only in far-field optical microscopy and nanoscopy but also in other cutting-edge applications based on fluorescence labeling, including single-molecule studies, (bio)analytical chemistry, and medical diagnostics.

## MATERIALS AND METHODS

### Synthesis of the rhodamine fluorophore KK 1518

The dye KK 1517, a previously described zwitterionic compound with a zero net charge [see Fig. 1 for the chemical structure and reference (9) for preparation and spectral data], was amidated with 1-methyl piperazine (a solubilizing alkylamine moiety) to give a corresponding product with a net charge of +2. The reaction was performed on a micromolar scale using O-(7-azabenzotriazol-1-yl)-*N,N,N′,N′*-tetramethyluronium-hexafluorophosphate (HATU) as a coupling reagent as follows: 3 mg (4 µmol) of the KK 1517 dye was dissolved in a mixture of dichloromethane (DCM; 2 mL) and dry acetonitrile

(MeCN; 2 mL) in a Schlenk flask. The flask was flushed with nitrogen and chilled to 5°C, and the following reagents were consecutively added under stirring and slow nitrogen purge: triethylamine (10 µL, 70 µmol), HATU (10 mg, 26 µmol), and 1-methyl piperazine (4 µL, 40 µmol). The solution was kept overnight at 4°C, subsequently diluted with DCM (5 mL), poured into an aqueous NaHCO$_3$ solution (0.1 M), and well shaken. The organic layer was washed with water (6 mL) and acidified with HCl (0.2 mL of a 37 wt.% solution), washed one more time with water, dried (Na$_2$SO$_4$ anhydrous), and evaporated. Purification was performed by means of repeated preparative high-performance liquid chromatography (HPLC) with a gradient of 40%–90% B (30 min) on a Kinetex EVO C-18 solid phase (column 21 × 250 mm) with A as aq. 0.025 M triethylammonium acetate (adjusted to pH 5) and MeCN as B. The absorbance detector was set to 635 nm, and pure fractions were pooled, concentrated to approximately 1/5 of the initial volume at $T$ ≤35°C, and combined with DCM (50 mL), followed by saturated brine (20 mL). The mixture was well shaken, and the organic layer separated, washed with water (10 mL), and acidified with HCl (0.5 mL) to remove triethylammonium salts from the organic phase. The organic layer was separated, the aqueous layer was extracted with DCM (~15 mL) until it became colorless, and the combined organic layers were washed with a 5 wt.% NaCl solution (20 mL), dried with Na$_2$SO$_4$ (anhydrous), filtered through a syringe filter with a pore size of 0.45 µm, and evaporated to give 2.8 mg (~80%) of the amide as a double chloride salt with MW = 895 (see Table 1 for structure). Later, the synthesis was repeated on a scale of 12 µmol with the same yield. The HPLC (Fig. S8a), liquid chromatography–mass spectrometry (electrospray ionization) [LC/MS (ESI), Fig. S8b], and nuclear magnetic resonance (NMR) spectroscopy (Fig. S9) confirmed the purity and identity of the compound. Analytical data for HPLC, thin-layer chromatography (TLC), LC/MS (ESI), and NMR spectra are provided in supplementary files.

## Synthesis of the rhodamine fluorophore KK 1116

The acid form of the KK 1115 dye with $q = -1$ [also known as "Abberior STAR 635," for the chemical structure, see references (7, 9), and Fig. 1] was amidated with dimethylamine to give a model dye with a zero net charge while keeping the solubilizing OH groups unreacted. The reaction was performed as described above and on the same scale (4 µmol). Dimethylamine was used as a 40% wt. aq. solution and was added at +5°C to the corresponding acid activated with HATU. After the workup of the reaction mixture, the product was isolated and purified by means of repeated preparative HPLC with a gradient of 10%–80% B (30 min) on a Kinetex EVO C-18 solid phase (column 21 × 250 mm) with A as aq. 0.025 M TEAB and MeCN as B. The absorbance detector was set to 635 nm, and pure fractions were evaporated and freeze-dried to give (~70%) the amide as zwitterionic salt. Later, the synthesis was repeated on a scale of 12 µmol with a similar yield. HPLC (Fig. S10a) and LC/MS (ESI) (Fig. S10b) analyses, as well as NMR spectra (Fig. S11), confirmed the purity and identity of the compound.

Analytical data for HPLC, TLC, LC/MS (ESI), and NMR are provided in supplementary files.

## Synthesis of the active *N*-hydroxysuccinimidyl ester of the oxazine fluorophore KK 1905

The oxazine-derivative KK 1905 [an unreactive compound with OH and carboxyethyl (CO$_2$Et) as functional groups; Fig. 1] was synthesized as previously described [see ref. (8) and compound 20-Et,H therein] and additionally purified by means of preparative HPLC with a gradient of 10%–90% B (30 min) on a Kinetex EVO C-18 solid phase (column 21 × 250 mm) with A as aq. 0.005 M trifluoroacetic acid and MeCN as B. The absorbance detector was set to 660 nm. Pure fractions were pooled and mixed with a double volume of saturated brine and DCM, respectively, and the pH was made slightly basic (7, 8) by adding a saturated aqueous NaHCO$_3$ solution. The mixture was well shaken, and the organic layer was separated, washed with a half-volume of 5 wt.% NaCl solution, dried with Na$_2$SO$_4$ (anhydrous), and evaporated under reduced pressure. In a previous

study, this dye was an intermediate compound employed for phosphorylation to give a well-performing STED fluorophore (8). Initially, attempts were made to hydrolyze the $CO_2Et$ group of the dye KK 1905 and then activate it *via* conventional NHS esterification. However, even a very weak (0.05 M) alkaline solution at 0°C caused the destruction of the dye core, an unexpected behavior completely different from its phosphorylated derivative, which hydrolyzed smoothly. Nevertheless, it was possible to prepare an NHS carbonate at the OH site of KK 1905 using the *N,N'*-disuccinimidyl carbonate (DSC) coupling reagent.

The active NHS ester (NHS carbonate) of the dye was obtained as follows: to a solution of KK 1905 (as a chloride salt with a free OH group, MW = 500.5, 10 mg, 0.02 mmol) in a mixture of DCM (5 mL) and dry MeCN (1 mL) was added DIPEA (35 µL, 0.20 mmol), a solid DSC coupling reagent (77 mg, 0.30 mmol), and 4-dimethylaminopyridine (2 mg, catalyst). The solution was stirred for 2 h at room temperature, and the reaction completion was witnessed by TLC and HPLC (see supplementary files for details). Then, the solution was diluted with DCM (40 mL) and quenched with a mixture of saturated brine (40 mL) and water (60 mL) containing trifluoracetic acid (100 µL) under vigorous shaking. The organic layer was washed twice with water (50 mL) to separate the *N*-hydroxysuccinimide (a product of DSC hydrolysis), dried with $Na_2SO_4$ (anhydrous), filtered through a syringe filter with a pore size of 0.45 µm, and evaporated at room temperature under vacuum. The NHS ester was thus isolated with a yield of 88% as a chloride salt with MW = 640.5. The residue (11 mg) was re-dissolved in dry *N,N*-dimethylformamide (1.10 mL), and the aliquots (containing 1 mg of the dye each) were evaporated in vials under vacuum as previously described (9) and stored at −20°C under a nitrogen atmosphere. Figure S12a depicts the synthesis of KK 1905-NHS from its precursor, KK 1905. The HPLC (Fig. S12b) and LC/MS (ESI) (Fig. S12c) analyses, as well as the NMR spectra (Fig. S13), confirmed the purity and identity of the compound. Only trace amounts of the starting compound KK 1905 and of other impurities were detected.

Analytical data for HPLC, TLC, LC/MS (ESI), and NMR are provided in supplementary files. The TLC of the three new compounds, KK 1518, KK 1116, and KK 1905-NHS, is shown in Fig. S14.

All the three dyes described above are dark blue solids sufficiently soluble in water and well soluble in most organic solvents to give blue solutions with a red fluorescence, which in the case of KK 1905-NHS is only slightly seen by a naked eye as its maximum is shifted to the IR region (see Results, Discussion, and data in Table 1).

## Quantification of the fluorescence relative quantum yields

The fluorescence relative quantum yield ($\Phi_f$) of the fluorophores was determined using the FluoroLog-3 spectrofluorometer (Horiba Scientific) equipped with an R2658 photomultiplier (Hamamatsu) with an excitation wavelength of 585 nm. Three measurements at different concentrations were performed for each dye sample (9). Maximum absorption was set below 0.05 to avoid an inner filter effect. $\Phi_f$ was determined using a solution of Oxazine 170 in ethanol as a reference dye ($\Phi_f$ = 0.579) and calculated as described elsewhere (60, 61).

## Bacterial strains and sample preparation for CLSM

The Gram-positive bacterium *Bacillus subtilis* subsp. *spizizenii* DSMZ347 and the Gram-negative bacterium *Escherichia coli* MG1655 were grown on Luria-Bertani agar (LA) plates, and three colonies of each species were suspended in LB and incubated at 37°C in shaking (i.e., 250 rpm). Incubation times were specified in the text for each experiment. After incubation, bacterial cells were harvested by centrifugation (3,000 × *g*, 5 min), washed, and diluted in sterile saline to reach an optical density at 600 nm ($OD_{600}$) of 1.0, which corresponds to approximately 2–5 × $10^8$ CFU/mL. An aliquot of each bacterial suspension was heat-inactivated by warming for 30 min at 100°C. Bacterial viability was evaluated by viable counts on LA plates. Untreated and heat-inactivated bacterial suspensions were stained using 10 µM of the synthesized dyes and incubated at room

temperature for 1 h in the dark. When specified in the text, bacterial cells were stained with 0.5, 10, and 60 µM of NR, SYTO 9, and PI, respectively. After staining, bacterial samples were centrifuged (3,000 × *g*, 5 min) and suspended in saline to remove the dye excess. Twenty microliters of each stained bacterial suspension was spotted on a glass coverslip covered with 0.5% (wt/vol) agarose to immobilize cells before imaging.

## Synthesis of liposomes and their preparation for CLSM imaging

Liposomes were prepared by the ethanol injection method employing POPC (Avanti Polar Lipids, Inc., Alabaster, AL, USA) as phospholipid (62). The POPC-liposomes were stained with 0.5 µM of NR, 10 µM of KK 1905-NHS, 10 µM of STAR RED (unreactive form; purchased from Abberrior and prepared as described elsewhere) (27), or a combination of 0.5 µM of NR and 10 µM of KK 1905-NHS. A 20-µL aliquot of the stained POPC-based liposomes was gently deposited on microscope glass slides, air dried at room temperature, and visually inspected by CLSM.

## POPC-based liposomes, DNA, and bovine serum albumin staining

The genomic DNA of *E. coli* MG 1655 was extracted using a QIAamp DNA Mini kit (Qiagen), according to the manufacturer's instructions. The fluorophores used in this work were added at a final concentration of 10 µM in POPC-based liposomes (5 mg/mL), BSA (purchased from Sigma-Aldrich; 1 mg/mL), and genomic DNA (100 ng/µL) solutions prepared in distilled water. One hundred microliters of these solutions was aliquoted into black, clear-bottom 96-well microtiter plates (Greiner), and the emitted fluorescence was measured using a Sparks 10M multilabel plate reader (Tecan), selecting the appropriate excitation and emission wavelengths for each molecule (Table 1). The autofluorescence of liposomes, BSA, and DNA was subtracted from each recorded emitted fluorescence. FFI was calculated by dividing the dye's fluorescence in the presence of POPC-based liposomes, BSA, or DNA, and the dye's autofluorescence (in the absence of the indicated macromolecules). Values of FFI above 1.2 (corresponding to 20% of the FFI increment) were considered predictive of fluorescence emission increases in the presence of liposomes, BSA, or DNA.

## CLSM, colocalization, and STED imaging

Stained bacterial cells and liposomes were visualized using the Nikon A1+ confocal laser scanning microscope equipped with an Apo TIRF 100× oil immersion objective (NA 1.49). Table S2 summarizes the CLSM setup parameters employed to optimize the signal/noise ratio for each bacterial sample. Images were acquired at a sampling dimension of 512 × 512 pixels and deconvoluted through the NIS-Elements Confocal deconvolution software using default parameters.

Whole-cell emission spectra were determined on *E. coli* heat-inactivated cells after proper staining with the indicated dye, using the Nikon A1+CLSM lambda scan function.

For NR and KK 1517, KK 1518, or KK 1905-NHS colocalization analysis, five images for each sample were acquired with a dimension of 512 × 512 pixels and were deconvoluted employing the NIS-Elements software, using default parameters. Spatial resolution plots, cytofluorograms, and percentages of colocalization area were determined using the ImageJ software (63). Pearson (64) and Manders (65) coefficients were calculated using the JACoP plugin of the ImageJ software, as described elsewhere (66).

Super-resolution STED microscopy imaging was carried out using a STEDYCON microscope (Abberior) equipped with a 100× oil immersion objective (1.4 NA). The imaging assays were performed using 640 and 775 nm lasers for excitation and STED depletion, respectively. Fluorescence was detected in the range of 650–730 nm. CLSM and STED images were acquired in tandem at a sampling dimension of 450 × 450 pixels and deconvoluted using the Huygens software (SVI, the Netherlands).

## Fluorophore toxicity evaluation and CLSM time-lapse experiment

*B. subtilis* and *E. coli* cells were inoculated in LB and incubated for 18 h at 37°C. Then, bacterial cells (approximately $10^5$ CFU/mL) were suspended in LB with or without adding 10 µM of the dyes, and $OD_{600}$ was monitored during time using a Sparks 10M multilabel plate reader (Tecan) for up to 16 h at 37°C.

*B. subtilis* has the capability to grow in a wider range of temperatures with respect to *E. coli* (67), without requiring specific incubation equipment for time-lapse experiments. Therefore, *B. subtilis* was used for live-cell imaging, and cells were preliminarily grown in LB for 48 h at 37°C (death phase). A glass coverslip, where 20 µL of bacteria (approximately $10^8$ CFU) was deposited, was covered with a thin layer (approximately 0.1 mm width) of LB 0.5% (wt/vol) agarose supplemented with 10 µM of KK 1905. Similarly, *B. subtilis* cells were stained with 10 µM of KK 1905-NHS for 1 h before spreading onto a thin layer (approximately 0.1 mm width) of LB 0.5% (wt/vol) agarose. Time-lapse fluorescence microscopy experiments were performed with the Nikon A1+ CLSM equipped with the Apo TIRF 100× oil immersion objective. Subsequently, 512 × 512 pixels of images were acquired and deconvoluted using the NIS-Elements Confocal deconvolution software with default parameters.

## Monitoring bacterial membrane turnover using two-step labeling

*B. subtilis* cells were inoculated in LB and incubated for 18 h at 37°C. Then, bacterial membranes were labeled using 0.5 µM of NR, and cells were suspended in LB at a final $OD_{600} = 0.5$ and incubated at 37°C in shaking (*i.e.*, 250 rpm). Immediately after suspension in LB ($t = 0$ h) and 0.5, 1, 2, and 4 h post-incubation, an aliquot of the bacterial suspension was drawn, and cell membranes were counterstained with 10 µM of KK 1905-NHS. Cells were washed with sterile saline twice to remove the excess dyes, deposited on a glass slide covered with 0.5% (wt/vol) agarose, and visualized with a Nikon A1+ CLSM equipped with the Apo TIRF 100× oil immersion objective. Images were acquired at a sampling dimension of 512 × 512 pixels and deconvoluted using the NIS-Elements Confocal deconvolution software. The fluorescence intensity of NR and KK 1905-NHS was quantified using a square ROI of 25 × 25 pixels centered on a single bacterial membrane. Then, the mean pixel intensity (MPI) for NR and KK 1905 was quantified using ImageJ, and the membrane staining contribution of the two fluorophores was estimated according to the following equations:

Eq. (1) $\text{NR membrane staining}\,(\%) = \left( \dfrac{\text{MPI}_{\text{NR}}}{\text{MPI}_{\text{KK 1905}-\text{NHS}}} \right) \times 100$

Eq. (2) $\text{KK 1905}-\text{NHS membrane staining}\,(\%) = 100 - \text{NR membrane staining}\,(\%)$

where $\text{MPI}_{\text{NR}}$ and $\text{MPI}_{\text{KK 1905-NHS}}$ represent the MPI calculated for NR and KK 1905-NHS, respectively. This procedure was applied to the membranes of 50 cells at each time point (i.e., 0, 0.5, 1, 2, and 4 h of incubation). Cells presenting a cytoplasm homogenously stained by KK 1905-NHS and NR were considered dead (see Results and Discussion) and were not included in the analysis.

The reverse two-step labeling experiment was performed with the same conditions but inverting NR with KK 1905-NHS staining.

## Statistical analysis

Statistical analysis was performed with the GraphPad Instat software (GraphPad Software, Inc., La Jolla, CA, USA). All data were analyzed using a two-tailed unpaired Student's *t*-test. Differences with a $P \leq 0.05$ were considered statistically significant.

## ACKNOWLEDGMENTS

We thank Dr. Alvaro Crevenna (Epigenetics and Neurobiology Unit, EMBL, Rome, Italy) for giving us the possibility to perform STED imaging and for his kind assistance.

This work was supported by the Excellence Departments grant (art. 1, comma 314–337 Legge 232/2016) to the Department of Science of Roma Tre University, PRIN 2017 (20177J5Y3P), both from MIUR, and GAVAP (A0375-2020-36558) from Regione Lazio. The authors acknowledge the support of NBFC for Roma Tre University (CN00000033) and Rome Technopole (F83B22000040006).

The funders had no role in study design, data collection, interpretation, or the decision to submit the work for publication.

M.L., G.C., and D.V. designed the experiments. K.K. developed, synthesized, and characterized the fluorophores. T.G. synthesized the POPC-based liposomes. M.L. acquired the microscopy data. M.L. and M.P. performed the data analysis. M.L. wrote the first manuscript draft, revised by G.R., T.G., D.V., P.V., G.C., and K.K. All authors had access to all the data, contributed to the revision of the manuscript, approved the final version of the manuscript, and had final responsibility for the decision to submit for publication.

## AUTHOR AFFILIATIONS

[1]Department of Science, Roma Tre University, Rome, Italy
[2]NBFC, National Biodiversity Future Center, Palermo, Italy
[3]IRCCS Fondazione Santa Lucia, Rome, Italy
[4]Department of Engineering, University Roma Tre, Rome, Italy
[5]glyXera GmbH, Magdeburg, Germany

## AUTHOR ORCIDs

Massimiliano Lucidi ⓘ http://orcid.org/0000-0003-3238-9164
Giordano Rampioni ⓘ http://orcid.org/0000-0002-1735-8565
Paolo Visca ⓘ http://orcid.org/0000-0002-6128-7039
Kirill Kolmakov ⓘ http://orcid.org/0000-0003-0837-7390

## AUTHOR CONTRIBUTIONS

Massimiliano Lucidi, Conceptualization, Data curation, Investigation, Methodology, Validation, Writing – original draft | Giulia Capecchi, Data curation | Daniela Visaggio, Conceptualization | Tecla Gasperi, Conceptualization, Data curation, Writing – review and editing | Miranda Parisi, Data curation | Gabriella Cincotti, Conceptualization, Writing – review and editing | Giordano Rampioni, Conceptualization, Supervision, Visualization, Writing – review and editing | Paolo Visca, Conceptualization, Funding acquisition | Kirill Kolmakov, Conceptualization, Methodology, Writing – review and editing

## DATA AVAILABILITY

The dyes used in this work are available on request to Dr. Kirill Kolmakov. Raw and processed data contributing to this study are available on reasonable request to the corresponding authors.

## ADDITIONAL FILES

The following material is available online.

### Supplemental Material

**Supplemental Data (Spectrum03690-23-s0001.pdf).** Figures S1 to S14; Tables S1 and S2; captions of Videos S1 and S2; general information for analytical data; analytical data for KK 1518, KK 1116, and KK 1905-NHS; supplemental references.
**Video S1 (Spectrum03690-23-s0002.mp4).** Time-lapse of *B. subtilis* cells harvested from the death phase and seeded on an LB 0.5% (wt/vol) agarose pad supplemented with KK 1905.

**Video S2 (Spectrum03690-23-s0003.mp4).** Time-lapse of *B. subtilis* cells harvested from the death phase, stained with KK 1905-NHS, and seeded on an LB 0.5% (wt/vol) agarose pad.

Open Peer Review

**PEER REVIEW HISTORY (review-history.pdf).** An accounting of the reviewer comments and feedback.

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
