## [Reviewer comments · Microbiology Spectrum]

Microbiology Spectrum

Expanding the microbiologist toolbox *via* new far-red emitting dyes suitable for bacterial imaging

Massimiliano Lucidi, Giulia Capecchi, Daniela Visaggio, Tecla Gasperi, Miranda Parisi, Gabriella Cincotti, Giordano Rampioni, Paolo Visca, and Kirill Kolmakov

Corresponding Author(s): Massimiliano Lucidi, Universita degli Studi Roma Tre Dipartimento di Scienze

Review Timeline:

Submission Date:	October 16, 2023
Editorial Decision:	November 7, 2023
Revision Received:	November 15, 2023
Accepted:	November 17, 2023

Editor: Philip Rather

Reviewer(s): Disclosure of reviewer identity is with reference to reviewer comments included in decision letter(s). The following individuals involved in review of your submission have agreed to reveal their identity: Stefan Stanciu (Reviewer #1); Minsu Kim (Reviewer #2)

Transaction Report:

DOI: <https://doi.org/10.1128/spectrum.03690-23>

Re: Spectrum03690-23 (Expanding the microbiologist toolbox *via* new far-red emitting dyes suitable for bacterial imaging)

Dear Dr. Lucidi,

Thank you for the privilege of reviewing your work. Below you will find the reviewer's comments. Both were supportive of your study, but one reviewer had comments that will need to be carefully addressed.

Revision Guidelines

Sincerely,
Philip Rather
Editor
Microbiology Spectrum

Reviewer #1 (Comments for the Author):

The work of Lucidi et al. aims to contribute to the field of super-resolved imaging of prokaryotic cells, by introducing the novel use of a series of photostable contrast agents emitting in the far-red suitable for STED nanoscopy of bacteria. In my opinion the work has been performed with care and rigor, and undoubtedly, this study can yield significant impact for the microbiology community. While I commend the authors for their fine work, I would like to express some recommendations on how to enhance the manuscript and some pending concerns:

Major comments:

1) Some stains that are not specifically designed/engineered for STED nanoscopy, behave well under the specific high beam power requirements of this technique (see DOI: 10.1038/s41598-022-17825-5). I am wondering if the authors have tested the dyes that are typically used for fluorescent imaging of bacteria such as PI or Syto 9, to see how they behave under STED conditions. To this end, it would be nice to add to the paper a figure showing the maximum resolution achievable by STED with a conventional dye routinely used for bacteria imaging (e.g. PI, Syto 9, or others) vs. the maximum resolution achievable by the newly proposed dyes. The authors should include as well a discussion focusing on what beam power conventional dyes bleach, vs. the beam power for which the STED-focused dyes bleach. Resolutions for such discussed cases should be benchmarked by FWHM comparison.

2) I think it would come in very handy for the reader if a table is added summarizing the use of conventional dyes for bacteria fluorescence imaging. This table can be accompanied by a discussion on current limitations/bottlenecks, that can potentially be circumvented by new dyes, such as those discussed in this paper, but not only.

3) The authors should better describe the complementarity of DIC and STED. They use this pair of modalities in many of their figures, without explaining in details why DIC was chosen for cross comparisons. They should also better highlight details not visible in DIC, but well visible in STED.

4) I don't really understand Fig. 4b. The authors should describe its content and meaning in more detail, and better highlight the take home message for this figure.

5) While super-resolved fluorescence microscopy is definitely a key family of techniques in bioimaging, such technique also face limitations (eg. See DOI: 10.1126/sciadv.aav8062) . It would be nice if the authors provide in the final part of the paper a brief perspective on the potential use of label-free super-resolved techniques (see DOI: 10.1002/lpor.202200029), not requiring contrast agents, for better understanding of prokaryotic species.

Minor comments:

-The diffraction limit ($\Delta x, \Delta y$) is generally acknowledged to stand as ~ 200 nm, while the authors state differently. The following sentence needs to be revised:

"The resolving power of conventional fluorescence microscopy depends on the wavelength of the fluorescence light and constraints on both the lateral ($\Delta x, \Delta y \sim 250$ nm) and axial ($\Delta z \sim 500$ nm) resolution.

-The sentence below is incorrect from many regards. The authors should check it and revise it:

"Fluorophores are selectively deactivated in the outer region reducing the emission from out-of-focus points using depletion lasers"

Reviewer #2 (Comments for the Author):

This article presents an interesting and useful test of various far-red dyes. The experiments are well designed and conducted. I do not have any suggestions.

The authors thank the Reviewers for their comments and constructive criticisms and provide responses in dedicated font, below.

Reviewer 1

The work of Lucidi *et al.* aims to contribute to the field of super-resolved imaging of prokaryotic cells, by introducing the novel use of a series of photostable contrast agents emitting in the far-red suitable for STED nanoscopy of bacteria. In my opinion the works has been performed with care and rigor, and undoubtedly, this study can yield significant impact for the microbiology community. While I commend the authors for their fine work, I would like to express some recommendations on how to enhance the manuscript and some pending concerns.

We thank the Reviewer for Her/His favorable comments on our work. When possible, the Reviewer's comments have been addressed to enhance the quality of the manuscript, as detailed below.

Major comments:

1) Some stains that are not specifically designed/engineered for STED nanoscopy, behave well under the specific high beam power requirements of this technique (see DOI: 10.1038/s41598-022-17825-5). I am wondering if the authors have tested the dyes that are typically used for fluorescent imaging of bacteria such as PI or Syto 9, to see how they behave under STED conditions. To this end, it would be nice to add to the paper a figure showing the maximum resolution achievable by STED with a conventional dye routinely used for bacteria imaging (*e.g.* PI, Syto 9, or others) *vs.* the maximum resolution achievable by the newly proposed dyes. The authors should include as well a discussion focusing on what beam power conventional dyes bleach, *vs.* the beam power for which the STED-focused dyes bleach. Resolutions for such discussed cases should be benchmarked by FWHM comparison.

The idea of exploring the potential use of conventional dyes typically employed for bacterial staining (e.g., PI and Syto 9) in STED microscopy is undeniably intriguing. However, to facilitate a meaningful comparison of STED imaging feasibility with different fluorophores, it is essential to choose dyes with similar excitation wavelengths to ensure that photostability assessment depends solely on varying the depletion laser power, and it is not influenced by the different laser used for fluorophore excitation. Unfortunately, at present we do not have access to a STED facility to conduct this kind of experiments. Please also consider that this investigation lies beyond the intended scope of our manuscript, which primarily focuses on the possible applicability of new FR dyes for live single-cell imaging of bacterial

components by means of confocal microscopy (CLSM). In the last part of the manuscript we present data showing that the KK 1905-NHS fluorophore can also be used for STED microscopy, but we did not have the possibility to compare its performance relative to other dyes previously used in STED analysis or additional dyes typically employed for bacterial staining by CLSM. This limitation has been declared in the amended version of the manuscript by adding a sentence that emphasizes the need for future comparative analysis aimed at investigating the performance in STED imaging of KK 1905-NHS relative to other fluorophores (lines 313-316).

2) I think it would come in very handy for the reader if a table is added summarizing the use of conventional dyes for bacteria fluorescence imaging. This table can be accompanied by a discussion on current limitations/bottlenecks, that can potentially be circumvented by new dyes, such as those discussed in this paper, but not only.

We thank the Reviewer for Her/His suggestion. A Table summarizing the applications of fluorophores commonly employed for bacterial imaging has been added in supplementary files (Table S1), and properly cited in the text (lines 82-83). As indicated by the Reviewer, possible limitations related to the use of conventional bacterial dyes have been already described in the manuscript (lines 74-81, 100-104) and are further discussed in some cited references (e.g., in ref. 23, 40, 60). Due to the complexity of the subject, we think that a detailed discussion of the limitations of each dye would be more appropriate for a review article and goes beyond the scope of this research article.

3) The authors should better describe the complementarity of DIC and STED. They use this pair of modalities in many of their figures, without explaining in details why DIC was chosen for cross comparisons. They should also better highlight details not visible in DIC, but well visible in STED.

Like all label-free techniques, the inability to tag specific bacterial components diminishes the informative content of DIC images. Therefore, DIC is typically combined with other techniques, such as confocal microscopy, to enhance the information obtained on bacterial morphology. That said, please let us clarify that DIC imaging has been applied only in combination with CLSM in this manuscript, and not with STED. The Nikon A1+ confocal microscope used in this work is equipped with a DIC module for this purpose. We routinely use this diffraction-limited technique to verify if all bacterial cells are labeled (Lucidi et al., 2018, doi: 10.1128/aac.02480-17; Lucidi et al., 2019, doi: 10.1128/AEM.01334-19; Lucidi et al., 2020, doi: 10.1002/jbio.202000097; Mellini et al., 2021, doi: 10.1128/AEM.02956-20). Furthermore, as mentioned in the manuscript (lines 144-149), DIC can serve as an additional method to gather information on bacterial viability. Unfortunately, the STED

microscope employed in this research (STEDYCON, Abberior) did not come equipped with a DIC module, preventing the simultaneous acquisition of DIC images with STED and the possibility of making a comparison between these two techniques. Additionally, as suggested by the Reviewer (see comment 5 below), a brief comparison between label-free and label-based approaches has been provided in the Conclusion section (lines 318-328).

4) I don't really understand Fig. 4b. The authors should describe its content and meaning in more detail, and better highlight the take home message for this figure.

Figure 4b has been more extensively described in the amended version of the manuscript according to the Reviewer's suggestion (lines 204-211: "Through tracing arbitrarily lines (depicted as dotted lines in Fig. 4a), the spatial resolution of NR and KK1517, KK 1518, or KK 1905-NHS was assessed by plotting fluorescence intensity values for each fluorophore against the line length. Leveraging NR's selectivity in labeling bacterial membranes (30), each maximum in the NR spatial resolution plots pinpointed a bacterial membrane (as denoted by black arrows in Fig. 4b). The extent to which the curves of KK1517, KK 1518, and KK 1905-NHS overlap with NR's curves provided a qualitative estimation of the selectivity of each fluorophore for binding to the bacterial membrane. Notably, only KK 1905-NHS spatial resolution plot was similar to that of NR, suggesting colocalization of these two dyes on bacterial membranes (Fig. 4b).").

5) While super-resolved fluorescence microscopy is definitely a key family of techniques in bioimaging, such technique also face limitations (eg. See DOI: 10.1126/sciadv.aav8062). It would be nice if the authors provide in the final part of the paper a brief perspective on the potential use of label-free super-resolved techniques (see DOI: 10.1002/lpor.202200029), not requiring contrast agents, for better understanding of prokaryotic species.

Following the Reviewer's suggestion, a brief discussion on label-free super-resolved techniques has been added in the Conclusion section (lines 318-329).

Minor comments:

The diffraction limit (xy) is generally acknowledged to stand as ~200 nm, while the authors state differently. The following sentence needs to be revised: "The resolving power of conventional fluorescence microscopy depends on the wavelength of the fluorescence light and constraints on both the lateral (xy ~ 250 nm) and axial (z ~ 500 nm) resolution.

We thank the Reviewer for raising this interesting point. The achievable lateral and axial resolutions of conventional microscopes are currently under debate. Several works set the maximum achievable xy and z resolutions at 250 nm and 500 nm, respectively (MacDonald et al., 2015 doi: 10.1007/978-1-4939-2309-0_19; Luo et al., 2016, doi: 10.1038/lisa.2016.60; Cosentino et al., 2019; doi: 10.1126/sciadv.aav8062), while others break these limits at 200 nm (xy) and 600 nm (xz) (Vicidomini et al., 2018, doi: 10.1038/nmeth.4593; Astratov et al., 2023, doi: 10.1002/lpor.202200029). Since no univocal definitions of lateral and axial resolutions are provided for diffraction-limited microscopy, we have modified the sentence as follows: “The resolving power of conventional fluorescence microscopy depends on the wavelength of the fluorescence light and constraints on both the lateral (xy ~ 200-250 nm) and axial (z ~ 500-600 nm) resolution.” (lines 92-94).

The sentence below is incorrect from many regards. The authors should check it and revise it: "Fluorophores are selectively deactivated in the outer region reducing the emission from out-of-focus points using depletion lasers".

We thank the Reviewer for Her/His constructive criticism. The sentence has been entirely rephrased, and the working principles of STED have been more elaborately explained in the text (lines 292-298).

Reviewer 2

This article presents an interesting and useful test of various far-red dyes. The experiments are well designed and conducted. I do not have any suggestions.

We express our gratitude to the Reviewer for appreciation of our work.

Re: Spectrum03690-23R1 (Expanding the microbiologist toolbox *via* new far-red emitting dyes suitable for bacterial imaging)

Dear Dr. Lucidi,

Your manuscript has been accepted, and I am forwarding it to the ASM production staff for publication. Your paper will first be checked to make sure all elements meet the technical requirements. ASM staff will contact you if anything needs to be revised before copyediting and production can begin. Otherwise, you will be notified when your proofs are ready to be viewed.

Sincerely,
Philip Rather
Editor
Microbiology Spectrum